DOI: 10.1038/s41467-017-01257-1　　**OPEN**

# Harnessing a catalytic lysine residue for the one-step preparation of homogeneous antibody-drug conjugates

Alex R. Nanna[1,2], Xiuling Li[1], Even Walseng[1], Lee Pedzisa[2], Rebecca S. Goydel[1], David Hymel[3], Terrence R. Burke Jr.[3], William R. Roush[2] & Christoph Rader[1]

Current strategies to produce homogeneous antibody-drug conjugates (ADCs) rely on mutations or inefficient conjugation chemistries. Here we present a strategy to produce site-specific ADCs using a highly reactive natural buried lysine embedded in a dual variable domain (DVD) format. This approach is mutation free and drug conjugation proceeds rapidly at neutral pH in a single step without removing any charges. The conjugation chemistry is highly robust, enabling the use of crude DVD for ADC preparation. In addition, this strategy affords the ability to precisely monitor the efficiency of drug conjugation with a catalytic assay. ADCs targeting HER2 were prepared and demonstrated to be highly potent and specific in vitro and in vivo. Furthermore, the modular DVD platform was used to prepare potent and specific ADCs targeting CD138 and CD79B, two clinically established targets overexpressed in multiple myeloma and non-Hodgkin lymphoma, respectively.

[1] Department of Immunology and Microbiology, The Scripps Research Institute, Jupiter, FL 33458, USA. [2] Department of Chemistry, The Scripps Research Institute, Jupiter, FL 33458, USA. [3] Chemical Biology Laboratory, Center for Cancer Research, National Cancer Institute, National Institutes of Health, Frederick, MD 21702, USA. Correspondence and requests for materials should be addressed to W.R.R. (email: roush@scripps.edu) or to C.R. (email: crader@scripps.edu)

Antibody-drug conjugates (ADCs) are emerging as one of the most promising classes of cancer therapeutics. The strategy of conjugating potent cytotoxic compounds to highly specific antibodies has already been proven effective through four FDA-approved and currently marketed ADCs, ~60 in clinical trials, and many others in preclinical development[1]. However, the majority of clinically translated ADCs rely on random conjugation to cysteine (Cys) or lysine (Lys) residues, resulting in a heterogeneous mixture of ADCs containing 0–8 drugs per antibody, each with distinct pharmacokinetic, efficacy, and toxicity properties[2]. For these reasons, several site-specific conjugation technologies have been developed for the preparation of homogeneous ADCs[3]. These strategies rely on chemical or enzymatic processes to conjugate a discreet number of drugs at distinct locations of the antibody. Chemical approaches typically use engineered carbohydrates[4–6] or engineered amino acids to provide an orthogonal handle for conjugation. One of the most advanced approaches is the thiomab technology, which involves mutating amino acid residues at particular sites in the constant domains of the antibody to Cys residues that can then be selectively conjugated via their thiol groups. This technology has been utilized to produce potent ADCs against a variety of targets and indications using different cytotoxic payloads[7–13]. Although a one-step engineered Cys conjugation platform has been developed recently[14], the most common strategy involves a series of reduction and oxidation steps. Similarly, the twenty-first natural amino acid selenocysteine with its reactive selenol group (selenomab technology[15]), other natural amino acids[16, 17], as well as unnatural amino acids that contain reactive keto[18–20] or azide[21, 22] groups have been incorporated into antibodies for drug attachment. Another set of strategies for generating site-specific ADCs builds on chemically or enzymatically converting N-terminal serine[23] or certain Cys residues in engineered peptide motifs[24], respectively, to aldehyde groups. A chemical approach that does not rely on engineered carbohydrates or engineered amino acids implements native Cys rebridging for selective conjugation[25–27]. Enzymatic approaches that rebuild glycans[28, 29] or utilize engineered peptide motifs[30–34] for generating site-

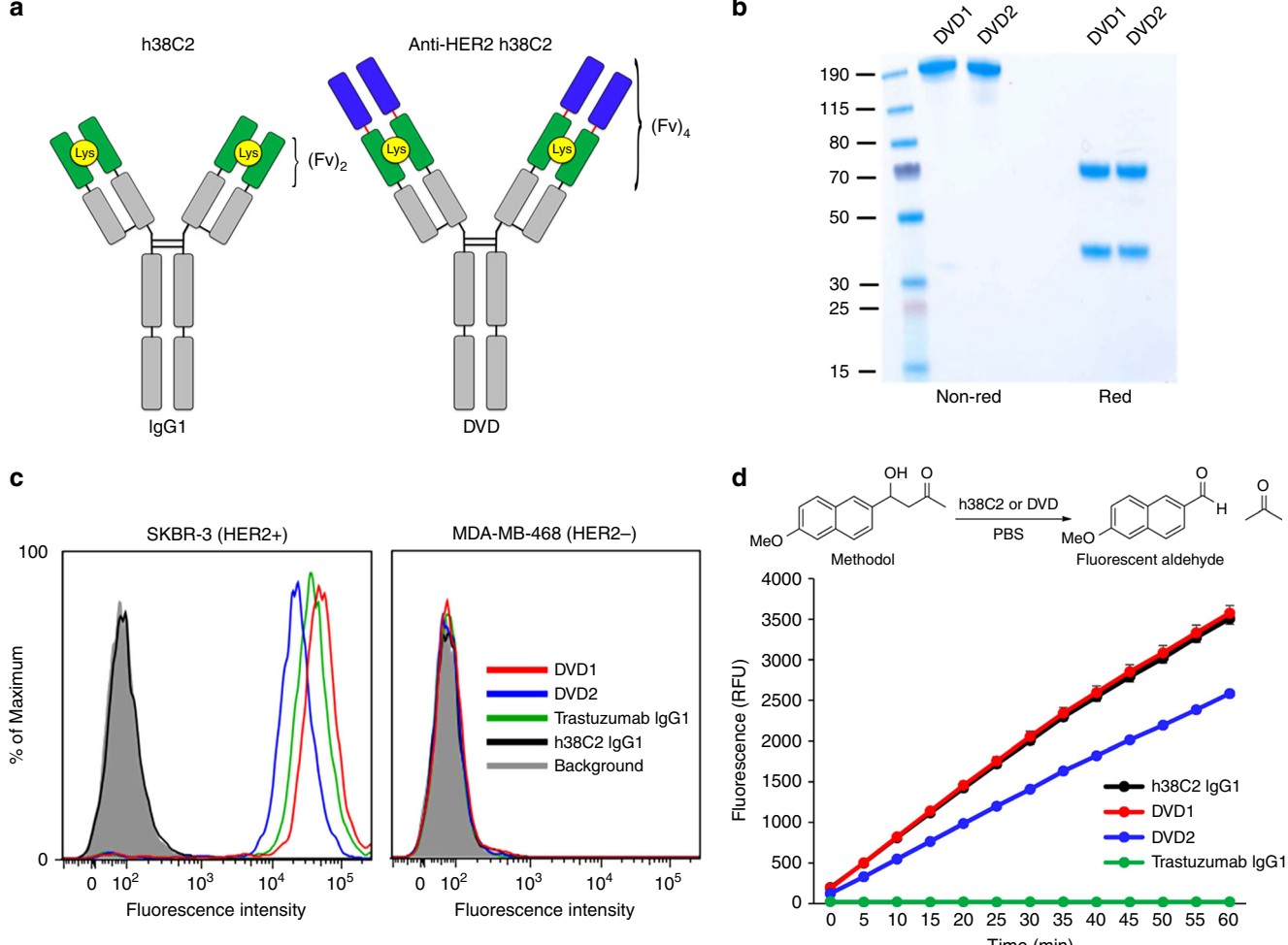

**Fig. 1** Construction of anti-HER2 DVDs. **a** h38C2 IgG1 compared to an anti-HER2 DVD. The DVD is composed of variable domains of trastuzumab (blue), h38C2 (green) with reactive Lys (yellow circle), and constant domains (gray). Two fully human spacer sequences (red lines) were used to prepare DVD1 with a short spacer (ASTKGP) or DVD2 with a long spacer (TVAAPSVFIFPP). **b** Coomassie stained SDS-PAGE confirmed the purity of both DVD1 and DVD2 under non-reducing (expected ~200 kDa) and reducing conditions (expected heavy chain ~63 kDa, light chain ~36 kDa). Molecular weights from a pre-stained protein ladder are shown on the left. **c** Flow cytometry showing specific binding of DVD1 (red) and DVD2 (blue) to SK-BR-3 cells (HER2+) with no binding detected to MDA-MB-468 cells (HER2−). Trastuzumab IgG1 (green) was used as a positive control and h38C2 IgG1 (black) as a negative control. **d** The catalytic retro-aldol activity of the reactive Lys of h38C2 was measured using methodol as a substrate. The signal is reported in relative fluorescent units (RFU; mean ± SD of triplicates). Trastuzumab IgG1 (green) was used as a negative control. The slope of DVD1 was not significantly different ($p = 0.1967$; linear regression analysis) from h38C2 IgG1, but the slope of DVD2 was significantly smaller ($p < 0.0001$)

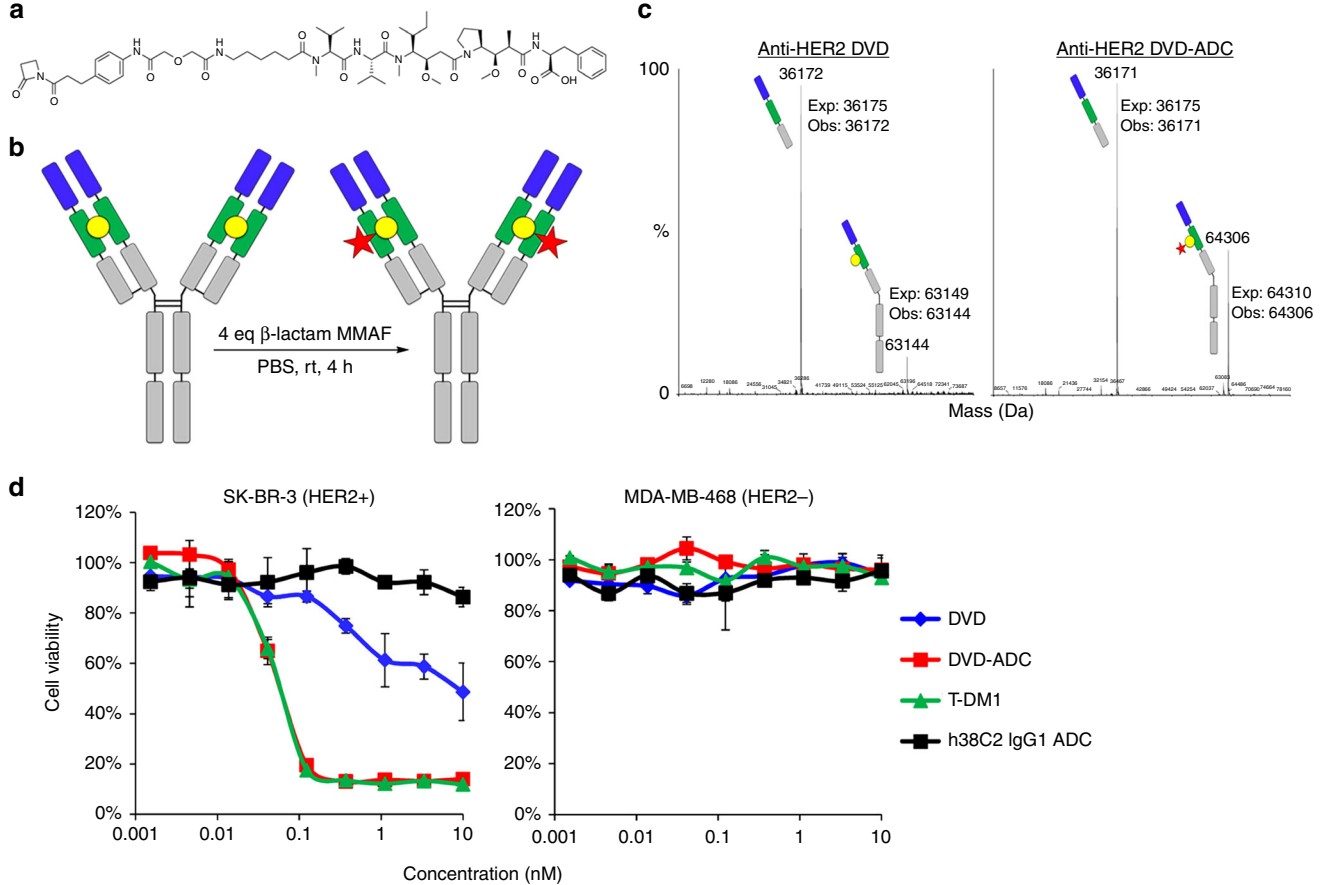

**Fig. 2** Assembly of DVD-ADCs and in vitro efficacy. **a** Structure of β-lactam MMAF. **b** anti-HER2 DVD-ADC was assembled by incubating the anti-HER2 DVD with four equivalents (eq) of β-lactam MMAF at room temperature (rt) for 4 h. Drug attachment (red star) occurs at the reactive Lys of h38C2 (yellow circle), located in the heavy chain variable domain of anti-HER2 DVD, to form a stable amide bond. **c** ESI-MS was performed with unconjugated anti-HER2 DVD and conjugated anti-HER2 DVD-ADC after reduction (10 mM DTT) and deglycosylation (PNGase F). No conjugation was detected on the light chain and the increase in mass of the heavy chain (~1160 Da) corresponds to the addition of exactly one eq of β-lactam MMAF. No unconjugated or higher drug loaded species were detected. **d** Cytotoxicity of anti-HER2 DVD-ADC (red) following incubation with HER2+ BC cell line SK-BR-3 and HER2− BC cell line MDA-MB-468 for 72 h at 37 °C (mean ± SD of triplicates). Unconjugated anti-HER2 DVD (blue) was significantly less toxic to SK-BR-3 cells when compared to anti-HER2 DVD-ADC ($p < 0.0001$; extra sum-of-squares $F$-test). T-DM1 (green) was used as a positive control and h38C2 IgG1 ADC (black, h38C2 IgG1 conjugated to β-lactam MMAF) as a negative control

specific ADCs have also been reported. Although existing site-specific ADC technologies have shown clear advantages over heterogeneous ADCs, an important consideration is that most of these strategies introduce mutations and it remains unclear if these will be problematic in immunocompetent patients. Furthermore, the mutation site can have drastic effects on the stability and efficacy of the resulting ADC[35], requiring that optimization be done for each antibody. In spite of the fact that recent advances have addressed stability issues[36] or developed alternative approaches as discussed above, most strategies still rely on mutations or inefficient conjugation chemistries often requiring long reaction times or multiple steps.

Lys conjugation is highly random with ~40 surface exposed Lys residues typically being available for conjugation[37]. However, we envisioned that we could harness the unique reactivity of h38C2, a humanized anti-hapten monoclonal antibody (mAb) containing a natural Lys at the bottom of an 11-Å deep hydrophobic pocket[38]. Due to its distinctive environment, this Lys is more nucleophilic (pKa ~6) at physiological pH, is capable of catalyzing aldol and retro-aldol reactions, and can be selectively conjugated to 1,3-diketone and β-lactam derivatives. The antibody has been used extensively to target antigens on cancer cells after being equipped with a targeting peptide, peptidomimetic, or other small

molecule using 1,3-diketone or β-lactam functionalized linkers. A variation of these h38C2-based chemically programmed antibodies[39] are chemically programmed bispecific antibodies that recruit and activate T cells[40]. The highly efficient conjugation reaction only requires an equimolar ratio of compound for complete covalent conjugation and does not modify other Lys residues due to the hydrophobic pocket being selective for hapten-derived compounds. These chemically programmed antibodies have also been shown to be well tolerated and do not elicit neutralizing immune responses in patients when assessed in phase I and II clinical trials[39]. Indeed, we postulated that use of h38C2 would allow the construction of a site-specific ADC platform having highly efficient, one-step conjugation chemistry that would not require mutations (i.e., no introduced amino acid residues for drug attachment), and would be, to the best of our knowledge, the first site-specific ADC to be generated using a natural Lys for conjugation.

## Results

**Engineering DVDs incorporating h38C2.** To harness the Lys reactivity of h38C2 for drug attachment and enable tumor targeting, we first engineered dual-variable-domains (DVDs)

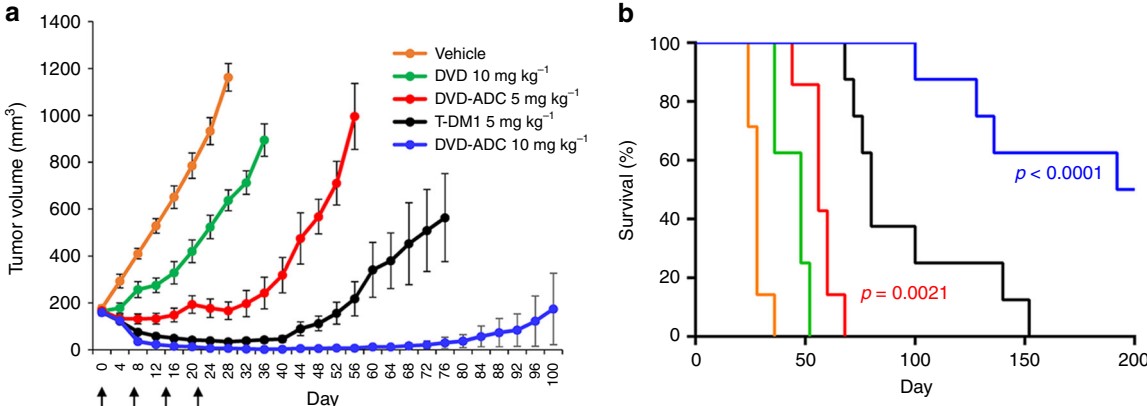

**Fig. 3** In vivo efficacy of anti-HER2 DVD-ADC. **a** Human BC cell line KPL-4 was xenografted into the mammary fat pads of female NSG mice, grown to ~150 mm³, randomized into five groups comprising seven or eight mice each, and treated with i.v. (tail vein) injections of the indicated ADCs and controls once a week for 4 weeks. The benchmark ADC, T-DM1 (black), was used as a positive control. Mean ± SEM values are plotted. When compared with a $t$-test (one-tailed, unpaired, uneven variance), the 5 mg kg⁻¹ (red) and 10 mg kg⁻¹ (blue) DVD-ADC groups were significantly different from the 10 mg kg⁻¹ DVD group (green) on day 8 ($p = 0.0046$ and $p = 0.00016$, respectively) until the last common data point on day 36 ($p = 0.0000069$ and $p = 0.0000019$, respectively). **b** Corresponding Kaplan−Meier survival curve with $p$-values (log-rank (Mantel-Cox) test) comparing survival between anti-HER2 DVD-ADC groups (red and blue) and the unconjugated anti-HER2 DVD group (green)

combining the variable domains of h38C2 and trastuzumab. DVDs are tetravalent IgG-like molecules composed of two heavy and two light chains. Despite their larger size (~50 kDa larger than the 150-kDa IgG), DVDs have similar pharmacokinetics and tissue penetration as IgG and retain a functional Fc domain for FcRn binding[41]. This format has also been clinically translated with two DVDs currently in phase II clinical trials[42]. For these reasons, we designed DVDs to enable our drug attachment strategy. Trastuzumab, the antibody component of FDA-approved ADC T-DM1 (ado-trastuzumab emtansine; Kadcyla)[43], was chosen for proof-of-concept studies. The constructed DVDs are essentially h38C2 IgG1 with the variable domain sequence of trastuzumab incorporated at the N-terminus (Fig. 1a). Thus, there are two HER2 binding sites and two drug attachment sites within one DVD molecule. Two different DVD constructs were generated that are identical except for the fully human spacer sequence[44] separating the h38C2 and trastuzumab variable domains. The DVDs were prepared in high purity (Fig. 1b) with retention of specific binding for HER2 (Fig. 1c). To ensure retention of the h38C2 Lys reactivity in DVD format, we used an assay directly assessing its catalytic activity through the conversion of methodol to its parent aldehyde (Fig. 1d) via a retro-aldol reaction[45]. DVD1 with the shorter spacer sequence was found to fully retain the catalytic activity of h38C2 IgG1. DVD2 with the longer spacer sequence was found to have diminished catalytic activity compared to h38C2 IgG1 and DVD1. Anticipating a correlation of catalytic activity and conjugation efficiency, DVD1 was pursued from this point on.

**Generation of anti-HER2 DVD-ADCs.** To prepare the desired ADC, a β-lactam functionalized monomethyl auristatin F (MMAF) compound with a non-cleavable linker was synthesized (Fig. 2a). The β-lactam handle was used for conjugation because it reacts irreversibly with the Lys by forming a stable amide bond[46], thus preventing the possibility of premature drug release. MMAF was chosen with a non-cleavable linker because this payload has been used to prepare potent ADCs against a variety of targets[47] and because ADCs with non-cleavable linkers have been reported to have higher maximum tolerated doses[48]. ADCs were prepared by incubating the DVD with four equivalents (eq) of β-lactam MMAF (two eq with respect to each Lys residue) in PBS for 4 h (Fig. 2b). As the Lys reacts with the β-lactam moiety to form an

amide bond, the Lys is no longer catalytically active, thus complete conjugation was confirmed by loss of catalytic activity (Supplementary Fig. 1). Mass spectrometry also confirmed the ADCs to be highly homogeneous, with only one drug attached per heavy chain (Fig. 2c and Supplementary Fig. 2), the location of the reactive Lys. Neither conjugation on the light chain nor higher drug loaded species were detected. Furthermore, when an alanine (Ala) was substituted for the Lys, the DVD was catalytically inactive (Supplementary Fig. 1) and no conjugation was detected after incubation with β-lactam MMAF (Supplementary Figs. 3, 4). These data combined with the fact that the DVD-ADC was no longer catalytically active, strongly suggest that the reactive Lys is the only conjugation point. As DVD-ADCs are composed of two heavy and light chains, the drug-to-antibody ratio (DAR) is 2.

**Validation of anti-HER2 DVD-ADC in vitro.** The in vitro cytotoxicity of the conjugate was evaluated against HER2+ (SK-BR-3) and HER2− (MDA-MB-468) breast cancer (BC) cells (Fig. 2d). The ADC was highly potent against target-expressing cells (IC₅₀ = 68 pM) with no cytotoxicity observed against the HER2− cell line. As a negative control, h38C2 IgG1 was incubated with β-lactam MMAF in parallel and tested against these cell lines. Since h38C2 lacks the additional anti-HER2 targeting domain, no cytotoxicity was observed as predicted. The ADC was also found to be highly potent against several other HER2+ cell lines and was as potent or superior to T-DM1 (Supplementary Fig. 5). Furthermore, the cytotoxicity was fully retained after incubation with human plasma at 37 °C for 3 days (Supplementary Fig. 6), indicating high linker stability of the DVD-ADC.

**Anti-CD138 and anti-CD79B DVD-ADCs.** To demonstrate broader utility of this strategy, two additional ADCs were prepared by substituting the HER2 targeting variable domain for a CD138 and CD79B targeting domain. Both of these antigens are clinically translated ADC targets for the treatment of multiple myeloma (MM)[49] and non-Hodgkin lymphoma (NHL)[50], respectively. The DVDs exhibited strong binding against target-expressing cells (Supplementary Fig. 7a) and were obtained in high purity (Supplementary Fig. 7b), with catalytic activity identical to the anti-HER2 DVD (Supplementary Fig. 7c). The desired DVD-ADCs were prepared using the same conditions

described for the anti-HER2 DVD-ADC (Fig. 2b). Both DVD-ADCs were catalytically inactive, indicating specific and complete conjugation at the desired Lys (Supplementary Fig. 7c). They were also highly potent against target-expressing cells with $IC_{50}$s in the double digit picomolar range (Supplementary Fig. 8).

**Validation of anti-HER2 DVD-ADC in vivo.** The in vivo activity was assessed using an orthotopic BC mouse model (HER2+ KPL-4 cells implanted subcutaneously in female NSG mice). Mice bearing established tumors (~150 mm$^3$) were treated every 7 days with an intravenous (i.v.) injection of 5 or 10 mg kg$^{-1}$ of anti-HER2 DVD-ADC, 10 mg kg$^{-1}$ unconjugated anti-HER2 DVD, the benchmark anti-HER2 ADC T-DM1 at 5 mg kg$^{-1}$, or vehicle for a total of four treatments. Since NSG mice lack serum IgG, each animal was pre-dosed with 250 µL of human serum to prevent clearance of the injected antibodies via Fcγ receptor-expressing phagocytes. Significant tumor regression and growth inhibition was observed for both anti-HER2 DVD-ADC doses (Fig. 3a and Supplementary Fig. 9). The survival was also significantly longer for both anti-HER2 DVD-ADC doses as compared to the unconjugated anti-HER2 DVD group (Fig. 3b). On day 100, the 10 mg kg$^{-1}$ anti-HER2 DVD-ADC group had seven out of eight animals surviving with five of these animals showing no tumor. At this dose, both tumor regression and survival were superior to T-DM1 at 5 mg kg$^{-1}$. Considering the size and drug loading of the DVD-ADC (200 kDa; DAR = 2) in comparison to T-DM1 (150 kDa; average DAR = 3.5)[51], the effective dose of cytotoxic agent of the DVD-ADC (~100 nmol kg$^{-1}$ MMAF) is lower than T-DM1 (~117 nmol kg$^{-1}$ mertansine). Despite this higher dose of active drug, no animals were cured during the course of the study using 5 mg kg$^{-1}$ T-DM1 and overall survival was shorter with only two out of eight animals surviving day 100 (Fig. 3b). Notably, half of the animals in the 10 mg kg$^{-1}$ anti-HER2 DVD-ADC group but no animal from the other four groups survived day 200 (Fig. 3b).

**Preparation of anti-HER2 DVD-ADC using crude antibody.** Inspired by the robust conjugation chemistry of our DVD-ADCs, we envisioned that drug conjugation could proceed using crude DVD samples. Indeed, we were able to produce desired DVD-ADCs by simply adding a solution of drug compound to crude DVD obtained after ammonium sulfate precipitation and dialysis. Despite the presence of other proteins, complete drug conjugation was indicated by loss of catalytic activity and the DVD-ADC retained potent activity in vitro (Supplementary Fig. 10). When compared to traditional purification methods, which involve at least two column chromatography steps, this present method only requires a single column chromatography step. In addition, our method is highly desirable due to its ability to rapidly prepare a panel of ADCs for testing. Furthermore, the efficient conjugation chemistry with this platform should enable higher throughput screening of ADCs[52] without a need for prior antibody purification.

## Discussion

In summary, we have developed a site-specific ADC platform that uses a natural Lys residue. The resulting ADCs are potent against several targets and have several key advantages over current formats. Most homogeneous ADC platforms introduce mutations in the constant domains of IgG1, which may increase the risk of immunogenicity[53]. The DVD format used in this strategy does not introduce any mutations in the constant domains and is a clinically translated format with two DVDs currently investigated in phase II clinical trials[42] and with h38C2 IgG1 previously investigated in phase I and II clinical trials[39].

Although the DVD format is unnatural, published clinical data with healthy volunteers report low antidrug antibody (ADA) responses[54]. A custom in silico T-cell epitope analysis (EpiMatrix)[55, 56] of the anti-HER2 DVD amino acid sequence revealed an overall low immunogenicity score (Supplementary Table 1). A linear regression analysis of FDA-approved and marketed mAbs with known immunogenicity estimates that an antibody candidate with this score triggers an ADA response in only 0.37% of patients. It should be noted, however, that this model was trained on mAbs with conventional IgG rather than DVD format. Also, the DVD light chain was predicted to be more immunogenic due to a higher score of the h38C2 variable light chain domain. We next analyzed the immunogenicity score of the junctional regions of the ASTKGP spacer between the two variable light and the two variable heavy chain domains and calculated a neutral score (Supplementary Table 2). Finally, because the drug component could act as a hapten once conjugated to the DVD, we assessed the immunogenicity of covalent modification of the reactive Lys with a small molecule by iteratively replacing Lys with all of the 19 remaining amino acids. None of these substitutions revealed a higher than neutral score (Supplementary Table 3). Collectively, while the anti-HER2 DVD reveals a low risk for immunogenicity overall, the variable light chain domain of h38C2, the junctional regions of the ASTKGP spacer, and the covalent modification of the reactive Lys indicate at least some risk for immunogenicity.

In ADC development, the conjugation site often has to be optimized to attain the desired stability and pharmacokinetics[31, 35]. By contrast, the DVD-ADC platform is based on a single point of attachment in a fixed inner Fv that is paired with an alterable outer Fv. Indeed, DVD-ADCs targeting HER2, CD138, and CD79B were shown to be highly potent against target-expressing cells. The stability of the conjugation point is also attributed to the inherent instability of the linker in many ADCs. For instance, the retro-Michael reaction of Cys based conjugates using maleimide drugs is well documented[57]. This process results in drug loss leading to a decrease in efficacy and increase in toxicity. Since the drug is the main driver of ADC toxicity in preclinical and clinical studies[58], drug conjugation must be stable. The h38C2 Lys reacts specifically with the β-lactam moiety to form a stabile amide bond, which is not labile even after incubation with human plasma. Moreover, since the reactive Lys is uncharged, drug conjugation does not eliminate any positive charges on the antibody. This is not the case with current Lys conjugation chemistries, which involve nucleophilic addition of the ε-amine of Lys with a succinimidyl ester, thereby eliminating a positive charge in the process and altering the electrostatic properties of the antibody[59]. Eliminating charges has implications for the stability, pharmacokinetics, and pharmacodynamics of therapeutic antibodies[60]. This conjugation strategy has also been shown to cause aggregation during the manufacturing of heterogeneous ADCs like T-DM1[61]. Thus, we conclude the reactive Lys in this platform has inherent advantages because it results in stabile, highly specific conjugation to Lys without eliminating a charge at the conjugation site.

The efficiency of drug conjugation is also an important consideration when developing ADCs. Ideally, the conjugation reaction is highly efficient, goes to completion in a short period of time, requires only stoichiometric amounts of drug, occurs in a single step without the need to buffer exchange, and does not require any reagents in addition to antibody and drug. Most site-specific ADC technologies do not meet these criteria. The Lys conjugation strategy we have developed uses two molar equivalents of drug with respect to each residue, is carried out in a single step using neutral conditions (i.e., PBS, pH 7.4), is complete in 4 h without the need to buffer exchange, and can be carried out using crude antibody preparations. Furthermore, a unique advantage of

this format is that drug conjugation can be measured directly by monitoring the reactivity of the reactive Lys using a catalytic assay. This analysis can be performed using purified ADC or crude conjugation reactions and only requires 1 h. Collectively, our method affords a uniquely robust, clean, and facile platform for making homogeneous ADCs.

## Methods

**Antibodies.** All variable domain sequences were based on published or patented amino acid sequences. All DVDs were prepared using amino acid sequences previously described[44]. Anti-HER2 DVDs were prepared by linking the VH and VL of trastuzumab to the VH and VL of h38C2 via a short (ASTKGP; the N-terminal 6 amino acids of human CH1) or long (TVAAPSVFIFPP; the N-terminal 12 amino acids of human Cκ) spacer for anti-HER2-DVD1 and anti-HER2 DVD2, respectively. The anti-HER2 DVD Ala mutant was prepared by substituting the Lys of DVD1 with an Ala using PCR. The anti-CD138 and anti-CD79B DVDs were expressed using the short (ASTKGP) linker. The desired sequences were synthesized as gBlocks (Integrated DNA Technologies) and expressed with a human IgG1 heavy chain or κ light chain constant domain. The DVD expression cassettes were NheI/XhoI-cloned into mammalian expression vector pCEP4 and transiently transfected into HEK293 cells cultured in DMEM medium supplemented with 10% fetal bovine serum (FBS), 100 μg mL$^{-1}$ streptomycin, and 100 U mL$^{-1}$ penicillin at 37 °C in an atmosphere of 5% $CO_2$ and 100% humidity. After 14–16 h, the media were discarded and replaced with expression medium (DMEM supplemented with 100 μg mL$^{-1}$ streptomycin and 100 U mL$^{-1}$ penicillin). The supernatants were collected three times over a 9-day period followed by filtration and purification using 1-mL HiTrap Protein A HP columns (GE Healthcare) in conjunction with an ÄKTA FPLC instrument (GE Healthcare). Yields were typically ~10 mg L$^{-1}$. The purity of DVDs was confirmed by SDS-PAGE followed by Coomassie staining, and the concentration was determined by measuring the absorbance at 280 nm. The complete amino acid sequences of the light and heavy chains of the anti-HER2, anti-CD79B, and anti-CD138 DVDs were published in international patent application *PCT/US2016/052214*. Control mAbs h38C2 IgG1 and trastuzumab were gifts from the laboratories of Drs. Carlos F. Barbas III (The Scripps Research Institute; La Jolla, CA) and Alfred Zippelius (University Hospital Basel; Basel, Switzerland), respectively. T-DM1 biosimilar was custom made by Levena Biopharma.

**Flow cytometry.** Adherent SK-BR-3 and MDA-MB-468 cells were collected using TrypLE (Life Technologies). U-266, NCI-H929, and Ramos cells grown in suspension were used directly. In a V-bottom 96-well plate (Corning), 100,000 cells per well were dispensed. The cells were washed with 200 μL flow cytometry buffer (PBS, 2% (v/v) FBS, 0.01% (w/v) NaN₃, pH 7.4), incubated with DVD or IgG1 (50 μL of a 20 nM solution in PBS) for 30 min on ice, washed with 200 μL ice-cold flow cytometry buffer, and stained for 20 min on ice with Alexa Fluor 647 conjugated polyclonal (Fab')₂ donkey anti-human IgG Fcγ fragment specific (Jackson ImmunoResearch Laboratories #709-606-098) diluted 1:100 in ice-cold flow cytometry buffer. After washing twice with 200 μL ice-cold flow cytometry buffer, the cells were analyzed using a Canto II Flow Cytometer (Becton-Dickinson). Data were analyzed using FlowJo software (Tree Star).

**Antibody conjugation.** All conjugations were performed in PBS (pH 7.4) after the antibodies were concentrated to 50 μM (10 mg kg$^{-1}$) using a 30-kDa cutoff centrifugal filter device (Millipore). Next, 6 μL of β-lactam-MMAF (1 mM in 10% (v/v) DMSO in PBS; 4 eq) was added to 300 μg of antibody for a final reaction volume of 36 μL. The solution was incubated for 4 h at room temperature (rt). All conjugations were deemed complete by loss of catalytic activity using the methodol assay for which a portion of the crude reaction diluted to 1 μM in PBS was used. Upon completion, unreacted compound was removed by using a PD-10 desalting column (GE Healthcare). To prepare ADC on a larger scale, 11.3 μL of β-lactam-MMAF (10 mM in DMSO) was added to 5.7 mg of antibody for a final reaction volume of 560 μL. The solution was incubated for 4 h at rt and purified by using a PD-10 desalting column as described above. The conjugates in PBS were stored at 4 °C for short term use and at −80 °C in aliquots for long term use. The concentration was determined by measuring the absorbance at 280 nm.

**Synthesis of β-lactam-MMAF.** The synthesis of β-lactam-MMAF is described in the Supplementary Methods. ¹H and ¹³C NMR spectra are provided as Supplementary Figs. 11–13.

**Catalytic activity assay.** Catalytic activity was analyzed using methodol[45, 62]. DVDs or IgG1s were diluted to 1 μM in PBS (pH 7.4) and dispensed in 98-μL aliquots into a 96-well plate in triplicate. Then, 2 μL of 10 mM methodol in ethanol was added and the fluorescence was assessed immediately using a SpectraMax M5 instrument (Molecular Devices) with SoftMax Pro software, a wavelength of excitation ($\lambda_{ext}$) set to 330 nm, a wavelength of emission ($\lambda_{em}$) set to 452 nm, and starting at 0 min using 5-min time points. The signal was determined by

normalizing to 98 μL PBS with 2 μL of the methodol solution added. The slopes were compared using linear regression analysis with GraphPad Prism Windows 6.01 software.

**Cell lines.** Human BC cell lines SK-BR-3, MDA-MB-468, and BT-474, human MM cell lines U-266 and NCI-H929, and human Burkitt lymphoma cell line Ramos were obtained from American Type Culture Collection (ATCC). (Note that cell line NCI-H929 is listed in Version 8.0 of ICLAC's Database of Cross-Contaminated or Misidentified Cell Lines (http://iclac.org/databases/cross-contaminations). However, its suspected misidentification[63] as a MM cell line by contamination with chronic myeloid leukemia (CML) cell line K562 was challenged[64]. We show in Supplementary Figs. 7, 8 that our NCI-H929 cells are strongly positive for CD138 and are selectively killed by our CD138-targeting DVD-ADC, respectively. In fact, they are indistinguishable in this regard from the other MM cell line, U-266, which we analyzed in parallel). Human BC cell line MDA-MB-361/DYT2 was kindly provided by Dr. Gregory P. Adams (Fox Chase Cancer Center; Philadelphia, PA) based on a Material Transfer Agreement (MTA) with Georgetown University (Washington, DC). Human BC cell line KPL-4[65] was kindly provided by Dr. Naoto T. Ueno based on an MTA with the University of Texas MD Anderson Cancer Center (Houston, TX) and with permission from Dr. Junichi Kurebayashi (Kawasaki Medical School; Kurashiki, Japan). An IMPACT III PCR Profile (IDEXX) revealed that both MDA-MB-361/DYT2 and KPL-4 cell lines were free of *Mycoplasma* and mouse viruses. BC cells were cultured in DMEM medium and MM and Burkitt lymphoma cells were cultured in RPMI 1640 medium, both supplemented with 10% FBS, 100 μg mL$^{-1}$ streptomycin, and 100 U mL$^{-1}$ penicillin at 37 °C in an atmosphere of 5% $CO_2$ and 100% humidity.

**In vitro cytotoxicity assay.** Cells were plated in 96-well plates at $5 \times 10^3$ cells per well for all BC cells, except KPL-4 ($3 \times 10^3$ cells per well). MM and Ramos cells were plated at $2 \times 10^4$ cells per well. BC cells were allowed to adhere overnight and suspension cells were treated immediately. Serial dilutions of unconjugated antibody and ADCs were added to the cells at concentrations ranging from 0 to 10 nM in half log steps. After incubation for 72 h, the cell viability was measured using the CellTiter 96 AQueous One Solution Cell Proliferation Assay (Promega) following the manufacturer's instructions. The cell viability was calculated as a percentage of untreated cells (≡100%). The IC₅₀ values were determined using logistic nonlinear regression analysis with GraphPad Prism Windows 6.01 software. The same software was used to determine p-values using an extra sum-of-squares F-test.

**In vitro plasma stability assay.** Anti-HER2 DVD-ADC was diluted to 1 μM using reconstituted human plasma (Sigma-Aldrich) (94% (v/v) human plasma in PBS). The sample was incubated at 37 °C for 72 h, then diluted to 10 nM using DMEM medium, and used for the cytotoxicity assay. Fresh anti-HER2 DVD-ADC was first diluted to 1 μM using reconstituted human plasma and then diluted to 10 nM using DMEM medium just prior to the cytotoxicity assay and used as a positive control. In vitro cytotoxicity was assessed using SK-BR-3 (HER2+) and MDA-MB-468 (HER2−) BC cell lines as described above.

**In vivo human breast cancer xenograft model.** KPL-4 cells ($6 \times 10^6$ per mouse) in a 1:1 mixture of PBS and BD Matrigel (BD Bioscience) were inoculated subcutaneously into the mammary fat pad of 7-week-old female NSG mice (The Jackson Laboratory). When tumors reached ~150 mm³, the mice were randomly assigned to five groups of 7–8 mice each and treated with anti-HER2 DVD-ADC at 5 mg kg$^{-1}$ ($n = 7$) or 10 mg kg$^{-1}$ ($n = 8$), or with unconjugated anti-HER2 DVD at 10 mg kg$^{-1}$ ($n = 8$), or with T-DM1 biosimilar (Levena Biopharma) at 5 mg kg$^{-1}$ ($n = 8$), or with vehicle (PBS) alone ($n = 7$), by i.v. (tail vein) injection every 7 days for a total of 4 cycles. The mice were pre-dosed with 250 μL of sterile-filtered human serum (Sigma-Aldrich) 24 h before each cycle by i.p. injection. The tumor size was monitored every 4 days using caliper measurement; p-values were determined with a t-test (one-tailed, unpaired, uneven variance) using Microsoft Excel 2013 software. Kaplan–Meier survival curve statistics were analyzed with a log-rank (Mantel-Cox) test using Graphpad Prism Windows 6.01 software. All procedures were approved by the Institutional Animal Care and Use Committee of The Scripps Research Institute and were performed according to the NIH guide for the care and use of laboratory animals.

**Mass spectrometry.** All samples were enzymatically deglycosylated with PNGase F (New England Biolabs) overnight at 37 °C under reducing conditions (50 mM DTT in PBS). The enzyme was removed using a Protein G HP SpinTrap (GE Healthcare) according to the manufacturer's instructions. The samples were reduced again prior to sample analysis (10 mM DTT in PBS). Data were obtained on an Agilent Electrospray Ionization Time of Flight (ESI-TOF) mass spectrometer. Deconvoluted masses were obtained using Agilent BioConfirm Software.

**Crude DVD preparation and conjugation.** To 100 mL of crude supernatant (DMEM supplemented with 100 μg mL$^{-1}$ streptomycin and 100 U mL$^{-1}$ penicillin) containing anti-HER2 DVD was added 44 mL of saturated (NH₄)₂SO₄ solution (to make 30% (NH₄)₂SO₄ as the final concentration) gradually while stirring at rt over

20 min. The solution was placed at 4 °C for 6 h and then centrifuged at 20,000×g for 20 min at 4 °C. The pellet was discarded and the supernatant was filtered through a 0.2-μm syringe filter. Next, 18.3 g of solid $(NH_4)_2SO_4$ (50% final salt concentration) was added gradually to the filtered supernatant at rt over 30 min while stirring, then stirred further for 15 min, and finally placed at 4 °C overnight. After 16 h, the solution was centrifuged at 20,000×g for 30 min at 4 °C. The pellet was suspended in 9 mL of PBS loaded into a 10-mL Float-A-Lyzer dialysis bag with a 100-kDa cutoff membrane (Spectrum Laboratories) and dialyzed against PBS at 4 °C overnight. The solution was then concentrated to 1 mL using a 100-kDa cutoff centrifugal filter device (Millipore). To 77 μL of crude DVD solution was added 4 μL of β-lactam-MMAF (10 mM in DMSO). The solution was incubated at rt for 4 h and purified by Protein A affinity chromatography as described above.

**In silico T-cell epitope analysis**. To assess the immunogenicity potential of the anti-HER2 DVD, a custom in silico T-cell epitope analysis was conducted by EpiVax (www.epivax.com) as follows:

T-cell epitope screening of heavy chain, light chain, and whole antibody: Heavy and light chain amino acid sequences were parsed into overlapping 9-mer frames and each frame was evaluated with respect to a panel of eight common class II human leukocyte antigen (HLA) alleles that cover ~95% of the human population[66]. Each frame-by-allele Assessment is a statement about predicted HLA binding affinity. EpiMatrix assessment scores range from approximately −3 to +3 and are normally distributed. Scores above 1.64 are defined as EpiMatrix Hits. These peptides have a significant chance of binding to HLA molecules with moderate to high affinity and, therefore, have a significant chance of being presented on the surface of antigen-presenting cells (APCs). In general, ~5% of all assessments can be expected to score above 1.64. All other factors being equal, the more HLA ligands (i.e., EpiMatrix Hits) contained in a given protein, the more likely that protein is to induce an immune response. The EpiMatrix Score is the difference between the number of predicted T-cell epitopes expected to be found in a protein of a given size and the number of EpiMatrix Hits. The EpiMatrix Score of an average protein is zero. Proteins with negative scores indicate the presence of fewer potential HLA ligands than expected and denote a lower potential for immunogenicity. Accordingly, proteins with positive scores indicate the presence of excess potential HLA ligands and higher potential for immunogenicity. Proteins scoring > 20 are considered to have a significant immunogenic potential. The Tregitope-Adjusted EpiMatrix Score accounts for the presence of tolerogenic T-cell epitopes, so called Tregitopes, which are found in mAbs[67]. To calculate the Tregitope-Adjusted EpiMatrix Score, which has been shown to be well correlated with observed clinical immune responses for a set of 23 FDA-approved and marketed mAbs[55], the scores of the Tregitopes are deducted from the EpiMatrix Score. Assessments, EpiMatrix Hits, Raw EpiMatrix Scores, and Tregitope-Adjusted EpiMatrix Scores for the anti-HER2 DVD are shown in Supplementary Table 1. The whole antibody had a Tregitope-Adjusted EpiMatrix Score of −34.29. A regression analysis trained on FDA-approved and marketed mAbs with known immunogenicity estimates that a mAb with this score will produce ADA in 0.37% of patients.

T-cell epitope screening of the junctional regions of the ASTKGP spacer: Although the ASTKGP spacer between the two variable light and the two variable heavy chain domains of the anti-HER2 DVD is a fully human sequence derived from the N-terminus of CH1, its junctional position between the dual-variable-domains is not naturally occurring and thus potentially immunogenic. To screen for T-cell epitopes in these two junctional regions, the flanking eight amino acids were added to the N-terminus and the C-terminus of ASTKGP (Cluster Sequences), parsed into overlapping 9-mer frames, and analyzed as described above to reveal EpiMatrix Hits and EpiMatrix Cluster Scores shown in Supplementary Table 2.

T-cell epitope screening of the Lys conjugation point: Covalent conjugation of small molecules to the reactive Lys can create immunogenic epitopes in the surrounding area. Indeed, small molecules have been shown to impact peptide binding in the context of class I and class II HLA[68–70]. While EpiMatrix does not model three-dimensional impacts for small molecules on HLA binding, it can assess the inherent binding potential of the linear amino acid sequence around the conjugation point. By iteratively replacing the wild-type conjugation point residue (Lys) with all of the 19 remaining amino acids, the degree to which the reactive Lys impacts HLA binding potential in its surrounding area gauges the effect of its covalent modification by a small molecule. Cluster Sequences, EpiMatrix Hits and EpiMatrix Cluster Scores for this sensitivity analysis are shown in Supplementary Table 3.

**Data reporting**. No statistical method was used to predetermine sample size, but sample size is similar to sample sizes routinely used in the field. The investigators were not blinded to allocation during experiments. No samples or animals were excluded.

**Data availability**. All data in this study are available within the article, its Supplementary Information, or from the corresponding authors upon request.

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

## Acknowledgements

We thank Rebecca F. Martin, William D. Martin, and Dr. Anne S. De Groot from EpiVax for conducting the in silico T-cell epitope analysis. We also thank Dr. Gogce Crynen for help with statistical analyses, and Drs. Gregory P. Adams, Naoto T. Ueno, and Junichi Kurebayashi for cell lines. Funding: We acknowledge support by NIH grant U01 CA174844 (C.R. and W.R.R.), the Klorfine Foundation (C.R.), and the Holm Charitable Trust (C.R.). This research was supported in part by the Intramural Research Program of the NIH, Center for Cancer Research, National Cancer Institute (T.R.B. Jr.). A.R.N. was the recipient of predoctoral fellowship awards from the Celia Lipton Farris and Victor W. Farris Foundation in 2015 and from the Division of Medicinal Chemistry (MEDI) of the American Chemical Society (ACS) in 2016.

## Author contributions

C.R. and W.R.R. conceived, designed, supervised, and analyzed all experiments; T.R.B. Jr. conceived, designed, analyzed, and supervised chemical syntheses; A.R.N. conceived,

designed, conducted, and analyzed chemical synthesis, biological syntheses, biochemical conjugations, and in vitro and in vivo experiments; X.L. conceived and supported in vitro and in vivo experiments; E.W. supported biological syntheses and biochemical conjugations; L.P. conceived, designed, conducted, and analyzed chemical syntheses; R.S.G. conduced biological syntheses, biochemical conjugations, and in vitro experiments; D.H. conducted chemical syntheses. A.R.N., C.R., and W.R.R. wrote the manuscript.

## Additional information

**Competing interests:** C.R., A.R.N., and W.R.R. are named inventors on international patent application *PCT/US2016/052214* claiming the DVD-ADC method. The remaining authors declare no competing financial interests. This is manuscript 29449 from The Scripps Research Institute.

