## [Peer Review File · Nature Communications]

Reviewers' comments:

Reviewer #1 (Remarks to the Author):

Reviewer's summary: This is a nice application of the already heavily-studied h38C2 antibody binding site, which has an unusually nucleophilic lysine inside. It bears the uncertainty associated with the relatively novel DVD fusion protein format, making in vivo applications in humans hopeful but uncertain. Overall the authors have done an excellent job of performing an important set of experiments to validate their strategy up to a preclinical level, and the experiments are worthy of publication. The main shortcomings are in the writing: this suggests that the more senior authors need to be more involved in producing the final version. The manuscript needs to be revised to focus on what is important and what has been accomplished, without overstating potential advantages for which little evidence is available.

Salient points:

1. The reader may not have the same confidence in the ultimate success of the DVD format that the authors have. The authors should provide more supporting evidence, or reexamine their approach to communicating these studies.
2. The repeated emphasis on advantages w/r immunogenicity is misplaced. The authors' approach has its own set of problems (e.g. the antigenic nature of the conjugate, distinct from that of the starting protein), which are not to be ignored. As long as there is a synthetic small molecule covalently attached to a protein, immunogenicity will need to be addressed. If there is a compelling justification for the authors' faith in the superiority of their approach, it is not communicated clearly in the current manuscript.
3. The authors have not cited a relevant recent paper from their lab (see below). Those skilled in the art will readily recognize the train of thought followed by their years of experiments with 38C2. It is better to keep the full picture in mind for the reader's benefit.

Further details:

Intro: The authors emphasize lack of mutations in their construct, but the dual variable domain molecule is an unnatural, engineered protein. Clearer presentation is needed. My understanding is that DVDs are still rather new, and one should not form conclusions about their immunogenicity just yet. There is also the more prominent issue that the immunoconjugate is likely to be immunogenic.

P5L88: I can't find the two spacer sequences, either in this ms or in ref 13. They should be readily accessible to the reader.

P10L184: More misleading words about immunogenicity. Refs 4 and 23 are focused on other questions. The authors should acknowledge the fact that successfully engineering a conjugation site into a constant region will permit multiple uses of that framework, analogous to using h38C2 in different DVD constructs.

Missing ref: THE JOURNAL OF BIOLOGICAL CHEMISTRY VOL. 291, NO. 37, pp. 19661–19673, September 9, 2016, by many of the same authors. Covalently attaches a cell-targeting hapten (folate) to the same h38C2 featured here, as part of a tumor-targeting diabody. One may argue that the scientific purpose is different, but the chemistry is the same, and the idea of attaching a small molecule useful for cancer therapy to h38C2 is foreshadowed by this article.

NOTE: Putting figure legends separate from figures is the bane of the reviewer's existence, forcing a hard-copy printout to avoid flipping between multiple display screens. Please don't do this.

Reviewer #2 (Remarks to the Author):

Whilst this paper is a good contribution to the field, I believe it lacks the novelty to be published in Nature Communications for the following reasons:

1. The IgG-DVD format is very well known and published on (several self-citations in the paper itself), and the b-lactam lysine chemistry modification for the scaffold used is also known and published on (see ref 15). The previously published IgG-DVD refer to complex bispecifics. The present work makes an IgG-DVD that is not a bispecific and where there is lysine modification with b-lactam on the same mAb scaffold used in ref 15. Thus the novelty can only be seen as going from a bispecific to a monospecific drug conjugate. A fail to see how this is a major advance, esp. in view of all the chemistry in ref. 15.
2. The introduction is very misleading - it is missing a large number of papers where native mAbs can be modified with site-selective chemistries (see multiple papers (>10) by Floris van Delft on glycan modification, Antony Godwin, Vijay Chudasama & David Jackson on disulfide modification, amongst many others). It gives the reader the wrong perspective of the work. Also there are more recent reviews on site-specific modification chemistries.
3. Having an antibody scaffold where it has previously been shown that it can be modified at a single lysine and doing the same thing again but adding a variable domain sequence at the N-terminus seems more complicated than modifying native mAbs using site-selective chemistries...or even using mutated mAb scaffolds..there should be more commentary in these comparisons
4. Characterisation - zoom ins for the mass specs are needed. Also needed are raw mass spec data - this is essential for NPG. NMR spectra should be shown.
5. Very confusing claims are made - " Furthermore, a unique advantage of this format is drug conjugation can be measured directly by monitoring the reactivity of the Lys using a catalytic assay. This analysis can be performed using purified ADC or crude conjugation reactions and only requires 1 h. No other ADC platform can directly measure the residue involved in drug attachment in this manner." - This is of course true but it is a statement about something specific to their format but people can take mass specs to see if the residues they are modifying or use Elmann's analysis etc.
6. Many other site-selective chemistries requires 2 molar equivalents (or even less) of reagent so not much novelty in this either.
7. Whilst "In summary, we have developed the first site-specific ADC platform using a natural Lys residue." is true, that same Lys residue has already been shown to be reactive using chemistry has already been optimised and it is all published.

Reviewer #3 (Remarks to the Author):

The authors describe an engineered ADC molecule that appears to have a remarkable set of properties, including homogeneous drug payload, free of induced mutations, stable bonding, no net change of charge, and single-step preparation.

I have a few comments.

1. The authors need to offer more discussion on why they chose DVD1 and eliminated the other construct. Was it merely because of the shorter spacer sequence?
2. Many of the figures contain curves in purple and dark blue, which are difficult to visually distinguish from one another.
3. A 2-tailed t-test in Fig 3A might be conservative (although it shows significance), since the context suggests a 1-sided test, i.e. DVD-ADC groups are less than DVD group.
4. In Fig 3B, I did not see any statistical assessment regarding difference of survival curves. DVD -

ADC (10mg/kg) seems visually evident, but difference between DVD (10mg/kg) and DVD-ADC (5mg/kg) is not as clear. Log-rank testing or some other appropriate statistical assessment would be helpful. Overall, statistical tests are lacking throughout the manuscript.

5. No legends for the supplementary figures.

Point-by-Point Response to the Reviewers' Comments

Reviewer #1

“This is a nice application of the already heavily-studied h38C2 antibody binding site, which has an unusually nucleophilic lysine inside. It bears the uncertainty associated with the relatively novel DVD fusion protein format, making in vivo applications in humans hopeful but uncertain. Overall the authors have done an excellent job of performing an important set of experiments to validate their strategy up to a preclinical level, and the experiments are worthy of publication. The main shortcomings are in the writing: this suggests that the more senior authors need to be more involved in producing the final version. The manuscript needs to be revised to focus on what is important and what has been accomplished, without overstating potential advantages for which little evidence is available.”

Response: We would like to thank the Reviewer for these kind words about our study and for suggesting improvements. We have carefully revised the text of our manuscript to shift the focus more toward our achievements rather than where we see opportunities for the DVD-ADC platform. To account for these and all other revisions, we have prepared an edited Main Text file with underlined changes.

“Salient points:

1. “The reader may not have the same confidence in the ultimate success of the DVD format that the authors have. The authors should provide more supporting evidence, or reexamine their approach to communicating these studies”.

Response: As mentioned above, we have revised the text to emphasize achievements over opportunities. In addition, as discussed below, we have carried out an extensive *in silico* T-cell epitope analysis to provide additional data in support of the therapeutic utility of the DVD-ADC platform.

2. “The repeated emphasis on advantages w/r immunogenicity is misplaced. The authors' approach has its own set of problems (e.g. the antigenic nature of the conjugate, distinct from that of the starting protein), which are not to be ignored. As long as there is a synthetic small molecule covalently attached to a protein, immunogenicity will need to be addressed. If there is a compelling justification for the authors' faith in the superiority of their approach, it is not communicated clearly in the current manuscript.”

Response: We agree with the reviewer that immunogenicity needs to be addressed. Although there are several methods to assess the immunogenicity of ADCs in the clinic (see, e.g., Hock, M. B. et al. Immunogenicity of antibody drug conjugates: bioanalytical methods and monitoring strategy for a novel therapeutic modality. AAPS J 17, 35-43 (2015)), nonclinical methods are limited. We therefore elected to perform custom *in silico* screening for T-cell epitopes using a state-of-the-art algorithm (EpiMatrix) developed by EpiVax (www.epivax.com). Computational T-cell epitope predictors have proven reasonably accurate in benchmark studies and have become an established strategy for gauging the immunogenicity of biotherapeutics and guiding their deimmunization (see, e.g., Griswold, K. E. & Bailey-Kellogg, C. Design and engineering of deimmunized biotherapeutics. Curr Opin Struct Biol 39, 79-88 (2016)). The EpiMatrix data for our anti-HER2 DVD are included in new Supplementary Figure 11. The overall immunogenicity score of the whole DVD-IgG1 was low (-34.29). A linear regression analysis of FDA-approved and marketed mAbs with known immunogenicity estimates that an antibody with this score will produce anti-drug antibodies (ADA) in 0.37% of patients. However, the

DVD light chain was predicted to be more immunogenic due to a high score for the h38C2 variable light chain domain (21.87). In addition to this analysis, the spacer sequence used to connect the two variable domains in the heavy and light chains was assessed and scored in the neutral range of the scale for potential immunogenicity. As pointed out by the Reviewer, covalent modification of the reactive Lys residue in the DVD-ADC has the potential to create new immunogenic epitopes. Thus, this site was screened for potential T-cell epitopes using a sensitivity analysis that assesses the potential of a given site to become immunogenic through mutation or covalent modification. The reactive Lys used for conjugation did not have any predicted T-cell epitopes and systematic substitution of this residue with the other 19 amino acids resulted in the formation of 0-4 T-cell epitopes, resulting in an overall neutral score for potential immunogenicity. Text was added to the Discussion (pages 10-11) to include this analysis along with two references (#55 and #56) describing the EpiMatrix scoring method and an additional reference (#54) addressing the immunogenicity of a clinically assessed DVD. The *in silico* T-cell epitope analysis was added to the Supplementary Methods (pages 14-15), Supplementary References (#4-#9), and new Supplementary Fig. 11 in the edited Supplementary Information file.

“3. The authors have not cited a relevant recent paper from their lab (see below). Those skilled in the art will readily recognize the train of thought followed by their years of experiments with 38C2. It is better to keep the full picture in mind for the reader’s benefit.”

Response: We thank the Reviewer for noting the absence of this source and agree that this study on utilizing h38C2 for chemically programmed bispecific antibodies should be mentioned. Text was added to include this reference (#40) on pages 4-5.

“Further details

4. “Intro: The authors emphasize lack of mutations in their construct, but the dual variable domain molecule is an unnatural, engineered protein. Clearer presentation is needed. My understanding is that DVDs are still rather new, and one should not form conclusions about their immunogenicity just yet. There is also the more prominent issue that the immunoconjugate is likely to be immunogenic.”

Response: This was addressed in our response to point 2 above.

5. “P5L88: I can’t find the two spacer sequences, either in this ms or in ref 13. They should be readily accessible to the reader.”

Response: This information was included in the Figure 1 legend of the original and revised manuscript: “Two fully human spacer sequences (red lines) were used to prepare DVD1 with a short spacer (ASTKGP) or DVD2 with a long spacer (TVAAPSVFIFPP).” It had also been included in the Methods section on page 12-13 “Anti-HER2 DVDs were prepared by linking the VH and VL of trastuzumab to the VH and VL of h38C2 via a short (ASTKGP; the N-terminal 6 amino acids of human CH1) or long (TVAAPSVFIFPP; the N-terminal 12 amino acids of human Ck) spacer for anti-HER2-DVD1 and anti-HER2 DVD2, respectively.”

6. “P10L184: More misleading words about immunogenicity. Refs 4 and 23 are focused on other questions. The authors should acknowledge the fact that successfully engineering a conjugation site into a constant region will permit multiple uses of that framework, analogous to using h38C2 in different DVD constructs.”

Response: We agree and modified this sentence on page 11 from *“In addition to the potential problem of immunogenicity, the site of the mutation often has to be optimized to attain the desired stability and pharmacokinetics. Since the point of conjugation is retained in the h38C2 variable domain, this platform can be used to target several antigens of interest without optimizing the point of attachment.”* to *“The conjugation site often has to be optimized to attain the desired stability and pharmacokinetics^{31,35}. By contrast, the DVD-ADC platform is based on a single point of attachment in a fixed inner Fv that is paired with an alterable outer Fv.”*

7. “Missing ref: THE JOURNAL OF BIOLOGICAL CHEMISTRY VOL. 291, NO. 37, pp. 19661–19673, September 9, 2016, by many of the same authors. Covalently attaches a cell-targeting hapten (folate) to the same h38C2 featured here, as part of a tumor-targeting diabody. One may argue that the scientific purpose is different, but the chemistry is the same, and the idea of attaching a small molecule useful for cancer therapy to h38C2 is foreshadowed by this article.”

Response: This was addressed in our response to point 3 above by adding the reference (#40) and corresponding text on pages 4-5. The reviewer is correctly pointing out that the scientific purpose (chemically programmed bispecific antibodies in diabody format vs. DVD-ADCs) is different but the conjugation chemistry is the same. Importantly, our current study is the first example of using a catalytic antibody to prepare an ADC and represents the first homogeneous ADC platform using a Lys residue. We would also like to point out that the catalytic activity of h38C2 is 100% preserved in the DVD format (revised Figure 1D) despite the fusion of additional variable domains at the N-termini of heavy and light chain. (See our response to Reviewer 3’s points 1 and 4). We found that this is not the case for several other bispecific antibody formats including the diabody format.

8. “NOTE: Putting figure legends separate from figures is the bane of the reviewer’s existence, forcing a hard-copy printout to avoid flipping between multiple display screens. Please don’t do this.”

Response: We apologize for separating the legends from the figures in the Supplementary Information. These figures are now paired with their corresponding legends in the edited Supplementary Information file.

Reviewer #2

“Whilst this paper is a good contribution to the field, I believe it lacks the novelty to be published in Nature Communications for the following reasons:

1. The IgG-DVD format is very well known and published on (several self-citations in the paper itself), and the b-lactam lysine chemistry modification for the scaffold used is also known and published on (see ref 15). The previously published IgG-DVD refer to complex bispecifics. The present work makes an IgG-DVD that is not a bispecific and where there is lysine modification with b-lactam on the same mAb scaffold used in ref 15. Thus the novelty can only be seen as going from a bispecific to a monospecific drug conjugate. A fail to see how this is a major advance, esp. in view of all the chemistry in ref. 15.”

Response: While the DVD format has been implemented previously to engage two antigens simultaneously, it has not been used for making ADCs. Specifically, the novelty lies in combining a targeting antibody (outer Fv) with a catalytic antibody for site-specific drug attachment (inner Fv). In fact, this is the first example of using a catalytic antibody

to prepare an ADC and represents the first homogeneous ADC platform using a Lys residue. Something that was not emphasized is that despite an entire variable domain being introduced at the N-termini of both heavy and light chain, the catalytic activity was 100% preserved. (See our response to Reviewer 1's point 7). This was confirmed by linear regression analysis. (See our response to Reviewer 3's points 1 and 4). Likewise, although the β -lactam conjugation chemistry has previously been reported, the conjugation efficiency is completely retained in the DVD format.

"2. The introduction is very misleading - it is missing a large number of papers where native mAbs can be modified with site-selective chemistries (see multiple papers (>10) by Floris van Delft on glycan modification, Antony Godwin, Vijay Chudasama & David Jackson on disulfide modification, amongst many others). It gives the reader the wrong perspective of the work. Also there are more recent reviews on site-specific modification chemistries."

Response: We agree with the Reviewer that other site-specific conjugation methods, in particular to native mAbs, were not amply discussed. New text and references (#4-#34) were added to include these strategies on pages 3-4. References using disulfide re-bridging by Godwin (#25), Chudasama (#26), and Jackson (#27) as well as glycan modification by van Delft (#28) and also by Boons (#29) were included. We also added a recent comprehensive review on ADCs in *Nature Reviews Drug Discovery* by Beck (#1).

"3. Having an antibody scaffold where it has previously been shown that it can be modified at a single lysine and doing the same thing again but adding a variable domain sequence at the N-terminus seems more complicated than modifying native mAbs using site-selective chemistries...or even using mutated mAb scaffolds...there should be more commentary in these comparisons."

Response: Please see our response to point 1 above.

"4. Characterisation - zoom ins for the mass specs are needed. Also needed are raw mass spec data - this is essential for NPG. NMR spectra should be shown."

Response: We agree with the Reviewer that this characterization should be included. New figures with non-deconvoluted and zoomed in deconvoluted ESI-MS spectra of the heavy and light chains were added (new Supplementary Figure 2 and new Supplementary Figure 4). Also added were the ^1H and ^{13}C NMR spectra for the reported shifts of compounds 3, 6, and 14 on pages 5, 7, 8, and 13 of the edited Supplementary Information file.

"5. Very confusing claims are made - "Furthermore, a unique advantage of this format is drug conjugation can be measured directly by monitoring the reactivity of the Lys using a catalytic assay. This analysis can be performed using purified ADC or crude conjugation reactions and only requires 1 h. No other ADC platform can directly measure the residue involved in drug attachment in this manner." - This is of course true but it is a statement about something specific to their format but people can take mass specs to see if the residues they are modifying or use Elmann's analysis etc."

Response: Our intention was to make the point that our catalytic assay is highly sensitive for measuring quantitative drug attachment even when using crude conjugation reactions. To avoid confusion, however, we removed the sentence "No other ADC platform can directly measure the residue involved in drug attachment in this manner".

“6. Many other site-selective chemistries requires 2 molar equivalents (or even less) of reagent so not much novelty in this either.”

Response: While other conjugation strategies also use nearly equimolar amounts of drug for conjugation, such as thiomab and cysteine re-bridging strategies, some enzymatic methods report a large excess. Nevertheless, we agree with the Reviewer and removed the corresponding text on page 12. However, we would like to emphasize that the novelty comes from the ease of conjugation in PBS (pH 7.4) in only 4 hours. Furthermore, the conjugation can be carried out using crude antibody samples as shown in revised Supplementary Figure 10.

“7. Whilst "In summary, we have developed the first site-specific ADC platform using a natural Lys residue." is true, that same Lys residue has already been shown to be reactive using chemistry has already been optimised and it is all published.”

Response: Please see our response to point 1 above.

Reviewer #3

“The authors describe an engineered ADC molecule that appears to have a remarkable set of properties, including homogeneous drug payload, free of induced mutations, stable bonding, no net change of charge, and single-step preparation.

I have a few comments.

1. The authors need to offer more discussion on why they chose DVD1 and eliminated the other construct. Was it merely because of the shorter spacer sequence?”

Response: We agree with the Reviewer that the reason DVD1 was pursued over DVD2 was inadequately discussed. Text was added on page 6 to the address this point. We also performed linear regression analysis to compare the slopes of DVD1 and DVD2 to h38C2 IgG1 from the catalytic assay (revised Figure 1D). There was no significant difference in the slope of DVD1 ($p = 0.1967$), but there was a significant decrease in the slope of DVD2 ($p < 0.0001$). This retention in slope, which indicates a fully conserved hapten-binding site, is the reason DVD1 was pursued over DVD2. This analysis was added to the legend of revised Figure 1D.

“2. Many of the figures contain curves in purple and dark blue, which are difficult to visually distinguish from one another.”

Response: We agree with the Reviewer and made color changes to the following figures for clarity and consistency: revised Figures 1C and 1D, revised Figure 2D, revised Supplementary Figure 1, revised Supplementary Figure 5, revised Supplementary Figure 7, and revised Supplementary Figure 8.

“3. A 2-tailed t-test in Fig 3A might be conservative (although it shows significance), since the context suggests a 1-sided test, i.e. DVD-ADC groups are less than DVD group.”

Response: We agree with the Reviewer that a two-tailed t-test is conservative and performed a one-sided t-test instead. The new p values (factor two lower) are now reported in the legend of revised Figure 3A.

“4. In Fig 3B, I did not see any statistical assessment regarding difference of survival curves. DVD-ADC (10mg/kg) seems visually evident, but difference between DVD (10mg/kg) and DVD-

ADC (5mg/kg) is not as clear. Log-rank testing or some other appropriate statistical assessment would be helpful. Overall, statistical tests are lacking throughout the manuscript.”

Response: We agree with the Reviewer and performed a Log-rank (Mantel-Cox) test to compare DVD-ADC (10 mg kg⁻¹) and DVD-ADC (5 mg kg⁻¹) groups to the DVD (10 mg kg⁻¹) group. Both the DVD-ADC (5 mg kg⁻¹) group and the DVD-ADC (10 mg kg⁻¹) group were found to be significantly different and the p values are reported in revised Figure 3B. We also agree with the Reviewer that additional statistical analyses would improve our manuscript. Thus, a biostatistician (Dr. Gogce C. Crynen, The Scripps Research Institute, Jupiter, FL) was consulted and several statistical analyses were added including linear regression analysis to compare catalytic activity slopes (revised Figure 1D, Supplementary Figure 7C) and an extra sum-of-squares F-test to compare IC₅₀ values (revised Figure 2D, revised Supplementary Figure 6, and revised Supplementary Figure 10B).

“5. No legends for the supplementary figures.”

Response: We apologize for separating the legends from the figures in the Supplementary Information. These figures are now paired with their corresponding legends in the edited Supplementary Information file.

REVIEWERS' COMMENTS:

Reviewer #3 (Remarks to the Author):

No additional comments.